# Metal and Metal Oxide Nanomaterials for Fighting Planktonic Bacteria and Biofilms: A Review Emphasizing on Mechanistic Aspects

**DOI:** 10.3390/ijms231911348

**Published:** 2022-09-26

**Authors:** Caixia Sun, Xiaobai Wang, Jianjun Dai, Yanmin Ju

**Affiliations:** 1College of Pharmacy, China Pharmaceutical University, Nanjing 211198, China; 2Department of Materials Application Research, AVIC Manufacturing Technology Institute, Beijing 100024, China; 3College of Life Science and Technology, China Pharmaceutical University, Nanjing 211198, China; 4Key Laboratory of Drug Quality Control and Pharmacovigilance (Ministry of Education), China Pharmaceutical University, Nanjing 211198, China; 5State Key Laboratory of Natural Medicine, China Pharmaceutical University, Nanjing 211198, China; 6MOE Joint International Research Laboratory of Animal Health and Food Safety, College of Veterinary Medicine, Nanjing Agricultural University, Nanjing 210095, China; 7Laboratory of Animal Bacteriology (Ministry of Agriculture), College of Veterinary Medicine, Nanjing Agricultural University, Nanjing 210095, China

**Keywords:** bacterial infection, metal and metal oxide nanomaterials, antibacterial, antibiofilm

## Abstract

The misuse and mismanagement of antibiotics have made the treatment of bacterial infections a challenge. This challenge is magnified when bacteria form biofilms, which can increase bacterial resistance up to 1000 times. It is desirable to develop anti-infective materials with antibacterial activity and no resistance to drugs. With the rapid development of nanotechnology, anti-infective strategies based on metal and metal oxide nanomaterials have been widely used in antibacterial and antibiofilm treatments. Here, this review expounds on the state-of-the-art applications of metal and metal oxide nanomaterials in bacterial infective diseases. A specific attention is given to the antibacterial mechanisms of metal and metal oxide nanomaterials, including disrupting cell membranes, damaging proteins, and nucleic acid. Moreover, a practical antibiofilm mechanism employing these metal and metal oxide nanomaterials is also introduced based on the composition of biofilm, including extracellular polymeric substance, quorum sensing, and bacteria. Finally, current challenges and future perspectives of metal and metal oxide nanomaterials in the anti-infective field are presented to facilitate their development and use.

## 1. Introduction

Bacterial infection with high morbidity and mortality rate have posed an increasingly serious threat to human health. The development of antibiotics has successfully resisted bacterial invasion. However, microorganisms have acquired drug resistance through genetic mutation or by receiving exogenous mutant genes because of the abuse of antibiotics [1,2]. They resist antibiotics by blocking drug–cell contact, rendering the drug inactive, altering their metabolic pathways, inhibiting the entry of drugs into the bacteria, or activating the efflux of drugs [2]. The era of antibiotics is progressively coming to an end due to the emergency of multidrug-resistant bacteria. It has been reported that multidrug-resistant bacteria cause at least 700,000 deaths worldwide each year, including 23,000 deaths in the United States and 25,000 deaths in the European Union [3]. The World Health Organization predicts that 10 million people worldwide will die from bacterial infections by 2050 if no efforts are made to reduce bacterial resistance or develop new antibiotics [4].

Infectious diseases caused by planktonic bacteria are common, such as sepsis and keratitis. When bacteria acquire drug resistance, removing the bacteria from the site of infection becomes a challenge. This challenge is amplified when bacteria form biofilm, which are one of the reasons bacteria develop drug resistance [5]. Bacterial biofilms are three-dimensional structure groups formed by bacteria embedded in their own secreted extracellular polymeric substance (EPS) [6,7,8]. Depending on the presence or absence of substrate during biofilm formation, they are classified into classical attached biofilms and type II nonattached biofilms. Biofilm formation is a cascading process, which can be divided into adhesion, reproduction, maturation, and shedding [9,10,11]. First, planktonic bacteria attach to the surface of the vector through outer membrane proteins and flagella, etc. Next, adhesion is enhanced by EPS secreted by the bacteria and which begins to develop into an irreversible adherent state by differentiation. They then develop into a mature three-dimensional structure through the aggregation of colonies. Finally, biofilms are hydrolyzed to release planktonic bacteria, which lead to the development of new infections. The formation of biofilm is beneficial to the survival of bacteria. The resistance capacity of biofilm reaches 1000 times that of planktonic bacteria. On the one hand, bacteria will undergo immune escape due to the protective effect of biofilm. On the other hand, the relevant components of the biofilm will interact with antibiotics to reduce their effective concentration. Furthermore, bacteria in biofilm can also resist the action of drugs through quorum sensing (QS) systems and dormant states [12,13,14]. Biofilms are commonly present on the surface of chronic wounds or implants, which can lead to delayed wound healing or implant removal [15]. The slow wound healing is attributed to decreased immune function, reduced cell proliferation, or reduced cellular reconstitution [16,17]. additionally, there are limited treatments available for implant infections, such as systemic antibiotic therapy or surgical treatment [12]. These will seriously affect the quality of life of the patient. Therefore, effective antibacterial and antibiofilm methods with no resistance are in urgent demand.

With the rapid development of nanotechnology, metal and metal oxide nanomaterials have emerged as promising therapeutics in anti-infective fields due to their excellent characteristics, including low cost and strong heat resistance. Importantly, they do not cause the development of drug resistance [18]. The anti-infective mechanisms of metal and metal oxide nanomaterials are mainly divided into three types, including the destruction of bacterial cytoskeleton, the generation of metal ions, and reactive oxygen species (ROS). They can not only kill planktonic bacteria, but also inhibit the formation of biofilm or destroy the formed biofilm by destroying EPS, inhibiting QS, and killing bacteria inside the biofilm. For example, He et al. constructed Au-Ag nanoshells to kill multidrug-resistant bacteria by high heat and silver ions (Ag^+^) [19]. Ali et al. pointed out that Au nanoparticles (NPs) can be used as nanoantibiotics to eliminate the biofilms of *Pseudomonas aeruginosa* [20]. From the aforementioned points, and taken together, metal and metal oxide nanomaterials provide a new weapon for the treatment of infective diseases.

It is essential to have a clear understanding of the antibacterial mechanism for developing new antibacterial materials. Compared to previous reports, this progress report concentrates on metal and metal oxide nanomaterials to summarize their anti-infective mechanism based on the components of bacteria and biofilms (Figure 1) [21,22,23,24]. We focus on the components of bacteria (cell membrane, protein, and nucleic acid) and conclude common antibacterial mechanisms of metal and metal oxide nanomaterials. In addition, we summarize relevant works based on metal and metal oxide nanomaterials for biofilm removal in terms of biofilm composition (EPS, QS, and bacteria). Ultimately, we also provide a detailed discussion of current challenges and future directions in metal and metal oxide nanomaterials for anti-infective applications. We believe that this progress report will provide new insights into the mechanisms of action of metal and metal oxide nanomaterials in antibacterial and antibiofilm applications and further facilitate their development and use.

## 2. Metal and Metal Oxide Nanomaterials

Metal and metal oxide nanomaterials refer to metals and alloys that form nanograins with small size effect, quantum effect, surface effect, and interface effect. They have unique physical and chemical properties compared to traditional metal and metal oxide materials, which have been widely investigated in recent years in different fields. Notably, they have been widely used in the anti-infective field due to their excellent anti-infective properties.

### 2.1. Metal Nanomaterials

Metal nanomaterials have been widely used in the field of anti-infection due to their excellent anti-infective efficacy. Among them, Ag, Au, and Cu are the most common anti-infective nanomaterial (Table 1). For thousands of years, the antibacterial properties of Ag have been discovered and used in everyday life, such as the use of silverware. The antibacterial properties of Ag nanomaterials mainly derive from Ag^+^ and depend largely on their size and shape. For example, Yang et al. used the displacement reaction between Zn and Ag^+^ to introduce Ag nanomaterials into the metal–organic framework. The antibacterial mechanism showed that massive release of Ag^+^ destroyed the bacterial contents and enhanced the effectiveness of the nanocomposite against *S. aureus* and *E. coli* [25]. Different from Ag nanomaterials, the antibacterial properties of Au nanomaterials are largely affected by their morphology and the antibacterial mechanisms are diverse. For example, Au NPs lack antibacterial activity on their own, but they showed favorable antibacterial activity by surface modification [26,27]. Based on this, Li et al. used 4,6-diamino-2-pyrimidinethiol modified Au NPs for the treatment of infections caused by *E. coli* [28]. In contrast, Au nanoclusters (NCs) with same surface ligand exhibit broad-spectrum antibacterial property by inducing the overaccumulation of ROS, not only against Gram-negative and Gram-positive bacteria, but also against their multidrug-resistant bacteria [29]. As another common anti-infective nanomaterial, Cu nanomaterials perform antibacterial properties mainly through generating ROS. On the one hand, Cu destroy the bacterial antioxidant system by causing the inactivation of glutathione reductase (GR), resulting in a surge of ROS [30]. On the other hand, cuprous ions originate from copper-based nanomaterials generated ROS due to Fenton-like activity. For example, Lin et al. constructed copper ion-loaded melanin and copper ion-loaded polydopamine to treat infective diseases induced by *S. aureus* and *E. coli* with the help of copper ion release and copper ion-induced ROS production [31]. Additionally, Pd and Pt nanomaterials can also act as nanozymes to produce ROS, resulting in broad-spectrum antibacterial activity against both Gram-negative and Gram-positive bacteria [32].

### 2.2. Metal Oxide Nanomaterials

Compared to metal nanomaterials, metal oxide nanomaterials have attracted the highest interest in the anti-infective community due to the better biological properties, such as TiO_2_ and ZnO (Table 2) [42]. As semiconductor nanomaterials, TiO_2_ and ZnO can generate highly toxic ROS by photocatalytic property, which are viewed as the promising tool for anti-infection therapy. However, their photocatalytic-based antibacterial properties were limited to the narrow response range of visible light and the easy recombination properties of photoinduced electron–hole pairs [43]. In order to improve their photocatalytic performance, combining with a semiconductor featuring narrow band gap has been reported. For example, Khan et al. reduced the band gap using Ag_2_S and decreased the rate of recombination for photoinduced charge carriers using graphene oxide (GO) [44]. The obtained Ag_2_S-ZnO/GO nanocomposite showed an outstanding photocatalytic property and remarkable antibacterial activity compared to pure ZnO nanomaterials. In addition, ZnO nanomaterials can also exert antibacterial properties through contact adsorption mechanism and metal ion dissolution mechanism. They all act by zinc ions derived from the degraded of ZnO nanomaterials in an acidic environment [45]. Zinc ions cause membrane potential disruption by adhering to the cell membrane. Moreover, they also act on the thiol group of the bacterial respiratory enzymes to increase the production of ROS, ultimately leading to bacterial death [46].

### 2.3. Synthetic Method

Different nanomaterials exhibit different degrees of anti-infective effect, which directly depend on their composition, morphology, and size. These characteristics are closely related to the synthetic method. There are various methods for the synthesis of metal and metal oxide nanomaterials, including chemical reduction, chemical precipitation, the Brust–Schiffrin method, seed-mediated growth, the hydrothermal reaction, and biosynthetic methods [52,53].

Chemical and biosynthetic methods have been reported for the preparation of metal-based nanomaterials. Among them, chemical reduction is the most common chemical method. However, the use of reducing agents raises the toxicity and cost of the method, as well as introduces impurities. As a simple and fast method, the chemical precipitation method has attracted a lot of attention, which allows for controlling the size and shape of the nanomaterials. The reaction rate and nucleation process are largely affected by reaction parameters such as pH, temperature, and reactant concentration. In addition, the size, shape, and properties of the nanomaterials will depend on the crystallization process. Based on this, Sondi et al. prepared well-dispersed Ag NPs with the size of 12.3 nm, using the reduction of AgNO_3_ by ascorbic acid [54]. Wang et al. synthesized morphologically controllable ZnO NPs nanoparticles with a particle size of about 20 nm using ZnCl_2_ as precursor and ammonium carbamate as precipitating agent [55]. As a nontoxic and environmentally friendly method, the biosynthetic method can be used to synthesize various nanomaterials with a wide range of size, physicochemical properties, shapes, and compositions, which has been widely reported. The sources of nanomaterials include plants, bacteria, and algae. For example, Mori et al. reported a simple and environmentally friendly route for the biosynthesis of Ag NPs [56]. The size of the Ag NPs was controlled by varying the glucose concentration. The final Ag NPs were produced with a controlled particle size range of 3.48 ± 1.83~20.0 ± 2.76 nm. Nasrollahzadeh et al. synthesized Cu NPs using the *Euphorbia grandis* leaf extract as a reducing and stabilizing agent without surfactants [52].

The Brust–Schiffrin method and the seed-mediated growth synthesis method can be employed to produce Au nanomaterials [57]. Brust–Schiffrin is the first special method that can generate thiolate-stabilized Au NPs. The Au NPs synthesized by this method have the following advantages: (1) high thermal and air stability, (2) no aggregation or decomposition occurring during repeated separation and redecomposition, (3) easy adjustment of the particle size of the synthesized gold nanoparticles with narrow dispersion range, (4) and relatively easy functionalization and ligand substitution modifications. For example, Selina Beatrice et al. prepared molecular tweezer-functionalized ultrafine Au NPs that selectively adsorb to lysine and arginine residues on protein surfaces using the Brust–Schiffrin method [58]. The process of the seed-mediated growth synthesis method of Au nanomaterials can be divided into two stages. First, small-sized Au NPs are synthesized as seeds. Second, the seeds are added to a “growth” solution consisting of HAuCl_4_, stabilizer, and a reducing agent. Then, Au^0^ produced by the reduction usually appears on the seed surface, and eventually a large amount of Au NPs is formed. Based on this method, MSc et al. obtained gold nanorods, gold nanostars, and gold nanospheres with a small size and a good dispersion [59].

The hydrothermal reaction is a method of synthesis using chemical reactions of substances in aqueous solutions at temperatures from 100 to 1000 °C and pressures from 1 MPa to 1 GPa. It can create new nanocompounds and nanomaterials that cannot be prepared by other methods because the homogeneous nucleation and nonhomogeneous nucleation mechanisms of the hydrothermal reaction are different from the diffusion mechanisms of the solid-phase reactions. Importantly, the products obtained by hydrothermal reaction have a high purity, good dispersion, and easy particle size control. For example, Wang et al. obtained well-dispersed copper nanowires using CuCl_2_·2H_2_O as the copper source and polyvinyl pyrrolidone as the dispersant. Ozga et al. used a modified hydrothermal reaction to obtain copper oxide films at less than 101 °C. Li et al. made ZnO nanoflowers by adjusting the ratio of zinc nitrate hexahydrate and cyclic hexamethylenetetramine [60]. Huang et al. prepared TiO_2_ nanotubes with high catalytic efficiency by the hydrothermal reaction.

To sum up, nanomaterials of the same component prepared by different methods may have different properties. It is important to choose an appropriate method by combining various factors.

## 3. Planktonic Bacteria

The main components of bacterial cells include cell walls, cell membranes, proteins, and nucleic acids. Among them, the cell membrane performs many functions in bacterial life, including energy metabolism, material transport, and the biosynthesis of bacterial structures. In addition, it is closely related to bacterial pathogenicity [61]. Protein is an important component of bacterial cells. All bacterial structures contain proteins, such as cell walls, cell membranes, ribosomes, etc. In short, protein is the main undertaker of bacterial life activities [62]. Nucleic acid is the genetic material of bacteria, which is used to store and express genetic information, including DNA and RNA [63]. Therefore, bacterial death can be achieved by the destruction of any of the above-mentioned components.

### 3.1. Cell Membrane

The bacterial cell membrane, which plays a selective barrier role, is a semipermeable membrane consisting of a phospholipid bilayer and proteins. The fundamental constituents of membrane proteins are amino acids, which can separate into positively charged amino groups and negatively charged carboxyl groups in solution. Thus, bacteria present negative charge under physiological conditions [64]. According to previous studies, metal and metal oxide nanomaterials can release metal ions under acidic conditions to induce bacterial death [65]. In particular, the electrostatic interaction between metal ions and bacterial cell membrane alters the membrane potential and causes K^+^ efflux, which further destroys the integrity of the cell membrane. The outflow of cytoplasmic contents through the pores eventually leads to bacterial death [54,66,67,68]. For instance, Nie et al. prepared Ag (−)/Ag (+) clusters with a positive zeta potential consisting of cysteine-modified Ag NPs and methyl maleic-acid-modified Ag NPs (Figure 2a). The results showed that their high bactericidal ability was attributed to the strong interaction between the positively charged nanoclusters and the negatively charged bacteria under the acidic environment [69]. Similarly, copper ions can combine with bacteria through electrostatic interaction to inhibit bacterial growth (Figure 2b,c) [70]. Moreover, zinc ions can also cause membrane potential disruption by adhering to the cell membrane. For example, Wu et al. proposed a win-win strategy based on a zinc sulfonate ligand–black phosphorus@ hydroxylapatite (ZnL_2_-BPs@HAP) for bacterial elimination and osteogenesis [71]. ZnL_2_-BPs showed a positively charged surface with the modification of ZnL_2_ (Figure 2d). The positively charged ZnL_2_-BPs disturbed bacterial membrane potential (Figure 2e,f).

Most of the time, the production of ROS is directly correlated with the release of cations. Large amounts of ROS will damage membrane proteins and lead to lipid peroxidation, ultimately causing collapse of the cell membrane structure [72]. Ma et al. constructed a pH-responsive oxygen and H_2_O_2_ self-supplying zeolitic imidazolate framework-67 (CaO_2_/GQDs@ZIF-67) nanosystem for photothermal therapy/chemodynamic therapy (CDT) combination antibacterial therapy (Figure 3a) [73]. The endogenous cobalt ion derived from ZIF-67 catalyzed H_2_O_2_ to produce hydroxyl radicals for CDT. The results of scanning electron microscope imaging and DNA and RNA leakage confirmed that severe cell membrane breakdown occurred (Figure 3b,c). Zhou et al. designed a metal–organic framework-based nanoplatform named MnFe_2_O_4_@MIL/Au&GOx for eliminating Gram-positive and Gram-negative bacteria (Figure 4a) [74]. It was found that MnFe_2_O_4_@MIL/Au&GOx could produce large amounts of ROS based on its peroxidase-like activity. Scanning electron microscope and transmission electron microscope images of *E. coli* and *S. aureus* demonstrated the cell-membrane-disrupting ability of ROS (Figure 4b).

The damage to cell membranes can be enhanced by combining metallic nanomaterials with nanomaterials possessing sharp borders [75]. For example, GO nanosheets with the sharp edges can break the lipid bilayer of the cell membrane and are often used in conjunction with metallic nanomaterials for synergistic antibacterial therapy [76]. Du et al. fabricated positively charged GO/nickel colloidal nanocrystal cluster (NCNC) nanocomposites by electrostatic self-assembly and exploited its synergistic antibacterial activity against *S. aureus* and *E. coli* [77]. Mechanistic studies have demonstrated that GO/NCNC nanocomposites could inhibit cellular growth by physically disrupting bacterial membranes. Notably, the antibacterial results showed that the antibacterial performance of GO/NCNC nanocomposites was higher than that of pure NCNC or GO suspensions (Figure 4c). Moreover, carbon nanosheets with sharp edge planes have also captured a lot of attention as potential antibacterial agents [78]. Sun et al. designed carbon nanosheets decorated with core–shell Cu@Cu_2_O nanoparticles (Cu@Cu_2_O/C-24) for antibacterial therapy [79]. The results showed that the sharp edges of carbon nanosheets and the produce of hydroxyl radical were responsible for bacterial death (Figure 4d,e).
Figure 4Cell membrane damage caused by reactive oxygen species and sharp edges. (**a**) The preparation process and antibacterial mechanism of MnFe_2_O_4_@MIL/Au&GOx (MMAG). (**b**) Transmission electron microscope image of bacterial morphology. White arrows represent serious damage of bacterial cells. (**c**) The bacterial morphological changes with different treatments assessed by scanning electron microscope. Red arrows represent serious damage of bacterial cells. GO, graphene oxide; NCNC, nickel colloidal nanocrystal cluster. (**d**) Evaluation of antibacterial properties of carbon nanosheets decorated with core-shell Cu@Cu_2_O nanoparticles (Cu@Cu_2_O/C-24). (**e**) Schematic diagram of the antibacterial mechanism of Cu@Cu_2_O/C-24 under the dark condition. (**a**,**b**) Reprinted with permission from Ref. [74]. Copyright 2022 American Chemical Society. (**c**) Reprinted with permission from Ref. [77]. Copyright 2022 American Chemical Society. (**d**,**e**) Reprinted with permission from Ref. [79]. Copyright 2021 American Chemical Society.
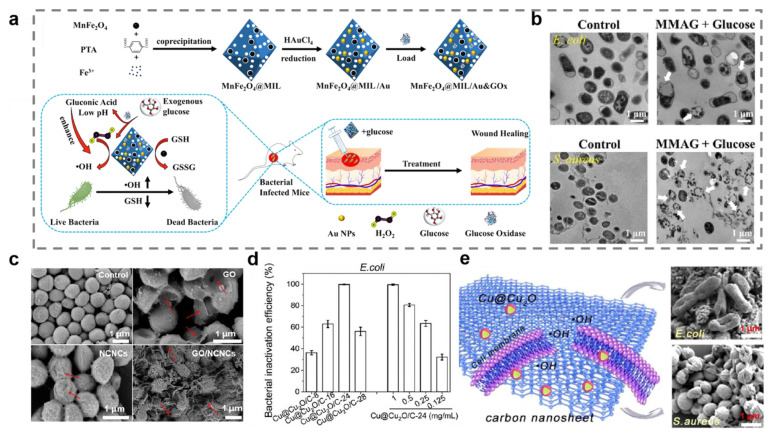


### 3.2. Protein

Instead of breaking cell membrane, metal and metal oxide nanomaterials can also eliminate bacteria by interfering with proteins. Aminoglycoside antibiotics work primarily by inhibiting the synthesis of bacterial proteins, which are mainly used for systemic infections caused by sensitive aerobic Gram-negative bacilli in clinical practice [80]. Inspired by the structural characteristics of aminoglycoside antibiotics, Wang and coworkers designed aminophenol modify Au nanoparticles (AP_Au NPs) with broad antibacterial spectrum against multidrug-resistant bacteria [81]. AP_Au NPs with hydroxyl groups bind to the 16S rRNA molecules of bacteria by forming hydrogen bonds to block bacterial protein synthesis, ultimately causing bacterial death (Figure 5). In addition to the direct inhibition of protein synthesis, metal and metal oxide nanomaterials can also break down natural enzymes that are essential for bacterial life, such as adenosine triphosphatases and GR [82]. The optimum temperature of natural enzyme is usually 37 °C. The enzyme activity will be diminished or even lost at high temperature due to the disruption of active sites [83]. Metal and metal oxide nanomaterials with magnetothermal or photothermal properties are capable of generating hyperthermia when treated with near-infrared light or alternating magnetic fields, which have drawn growing interest in the antibacterial field [84,85]. Sun et al. constructed a unique yolk–shell Fe_2_C@Fe_3_O_4_-PEG thermogenic nanozyme with highly magnetothermal properties for synergistic antibacterial application [86]. This nanozyme could generate hyperthermia with the treatment of alternating magnetic fields. The results of the enzyme activity assay revealed a nearly 80% inhibition of the typical natural enzyme activities in bacteria (Figure 6a,b). Copper is able to bind to GR to induce its inactivation, according to the previous study [87,88]. GR can catalyze the reduction of glutathione disulfide (GSSG) to reduced glutathione (GSH) to maintain redox homeostasis in bacteria [43]. If the activity of GR is inhibited, the cellular antioxidant system will be imbalanced and the bacterial cell will be apoptotic due to excessive intracellular ROS. Zhang et al. showed novel antibacterial copper clusters (CuCs) with excellent broad-spectrum antibacterial activity [30]. The results indicated that CuCs could not only disassemble bacterial membranes, but also provide to the imbalance of the GSH/GSSG ratio via suppressing the activity of GR, eventually leading to bacterial death (Figure 6c,d). Apart from Cu and Fe nanomaterials, ultrasmall Au nanoclusters can promote the upregulation of pro-oxidase and inhibit reductase, causing bacterial death by inducing metabolic imbalance of bacterial cells [89]. In addition, the small-sized Ag NPs can penetrate the bacterial cell membrane to cause intracellular enzyme inactivation through reacting with the -SH of enzyme [90]. Ag NPs can also inhibit signal transduction pathways of bacteria to eradicate bacteria [91].

### 3.3. Nucleic Acid

In addition to the interference of proteins by metal and metal oxide nanomaterials, another antibacterial strategy is the nanomaterial-induced inhibition of gene expression. The two-component signaling system (TCS) enables bacteria to sense, respond, and adapt to a wide range of environments, stressors, and growth conditions, which is closely linked to bacterial metabolism and virulence [92]. The tricarboxylic acid cycle (TCA) is the most efficient way for bacteria to obtain energy from the oxidation of sugars or other substances [93]. On this basis, Tu and coworkers prepared a hyperbranched poly-L-lysine (HBPL)-crosslinked hydrogel containing MnO_2_ for the therapy of infected diabetic wound (Figure 7a) [94]. The transcriptome analysis showed that most genes encoding pathogenesis-associated toxins and regulating TCA were downregulated after treating with HBPL (Figure 7b–f). The HBPL-modified MnO_2_ nanosheets could convert excess ROS to oxygen. The hydrogel dressing can consume multiple ROS to produce O_2_ and promote NO synthesis, ultimately killing MRSA effectively and achieving scarless wound healing. The upregulation of TCA-associated genes also contributes to bacterial clearance. According to reports, the upregulation of TCA-related genes restored bacterial susceptibility and enhanced the efficacy of antibiotics [95]. Li and coworkers constructed Ag nanoflowers to tackle the antibiotic resistance crisis with small doses of antibiotics and Ag nanoflowers [96]. The study found that Ag^+^ could disrupt cell membranes by inducing compensatory respiration in bacteria to produce more ROS, which eventually caused the entry of Ag^+^ and antibiotics into the bacterial cells (Figure 8 and Figure 9). Moreover, bacterial susceptibility was restored by the activation of the TCA cycle. Bacteria were finally cleared by minor doses of antibiotics and Ag nanoflowers. In addition, Ag^+^ can prevent DNA replication through altering the structure of DNA molecules of bacteria from an unfolded form to a folded state [97]. Ag^+^ can also trigger DNA mutations because guanine N7 and adenine N7 in DNA are the appropriate binding sites for Ag^+^ [98].

## 4. Biofilm

Bacterial biofilms are organized groups of bacteria wrapped in EPS on the surface of living or nonliving objects [9]. It has been shown that EPS occupies about 2/3 of the volume of the biofilm [99]. It connects thousands of bacteria together and provides a protective barrier against harsh environments, antibiotics, and the host immune system. The organization of biofilm derives from the QS of bacteria [100]. It is one of the main regulatory mechanisms of bacterial life activity and can coordinate the behavior of colonies. Interference with QS has the potential to prevent the pathogenicity of harmful bacteria. As another component of biofilm, internal bacteria are responsible for the pathogenicity of biofilm. Therefore, the removal of biofilm can be started by destroying EPS, QS, and killing internal bacteria.

### 4.1. Extracellular Polymeric Substance

As a direct environment for bacterial growth, EPS is one of the most crucial tools employed by microorganisms to defend cells against harmful external factors [101]. By preventing the contact of nanomaterials with bacteria, EPS, which is mostly composed of polysaccharides, proteins, nucleic acids, and lipids, may lessen the antibacterial effectiveness of nanomaterials [102]. It has been proven that destroying EPS is one of the most effective approaches to remove biofilm. An extensive interest in the field of antibiofilm has been shown for magnetic-field-controlled swarming of micro/nanorobots with high cargo loading and strong convection capabilities in swarm movement [103]. They can not only destroy EPS by physical movement, but also facilitate the penetration of a loaded antibacterial agent into the deeper biofilm. Ferrites, which exhibit persistent paramagnetism, are frequently employed in the production of various magnetic micro- and nanorobots, including Fe_3_O_4_ and Fe_2_O_3_ nanomaterials [104]. The swarm motion of micro/nanorobots broke EPS by generating strong convection and mechanical force in the presence of an external magnetic field. At the same time, iron-based nanomaterials were capable of generating ROS based on intrinsic peroxidase-mimicking properties for scavenging EPS and killing bacterial cells [104]. The physical destruction of EPS promoted ROS to penetrate the biofilm deeply, enhancing the antibiofilm effect of nanomaterials (Figure 10a,b) [105]. Moreover, micro/nanorobots can also be equipped with photosensitizers to improve the effectiveness of photodynamic disinfection. For example, Balhaddad et al. constructed a photosensitizer nanoplatform by integrating toluidine-blue ortho photosensitizer and Fe_2_O_3_ NPs (Figure 10c). In the action of an external magnetic field, the nanorobot-loaded photosensitizer entered the deep region of the biofilm. The effect of the photodynamic therapy was improved by the direct interaction between the photosensitizer and the bacteria (Figure 10d) [106]. It has been reported that EPS elimination may result from the degradation of EPS components. Environmental DNA (eDNA) can be used as a source of nutrients for bacteria in starvation environments, as well as bacterial intercellular linker and adhesion element [101]. The EPS structure will be lost if the eDNA is cleared [107,108]. Xia et al. used gallium to disperse the mature biofilms in an eDNA-dependent manner and transform it to the immature biofilm state [109]. The findings indicated that mature biofilms treated with gallium have reduced degrees of antibiotic tolerance. Coadministration of vancomycin would eliminate the biofilm (Figure 10e).

### 4.2. Quorum Sensing

QS is a way of communication between bacteria that relies on autoinducers. Autoinducers are synthesized by bacteria and released into the surrounding environment for intra- and interspecies conversation [110]. Gram-negative bacteria generally use acylhomoserine lactone-type molecules as autoinducers, while Gram-positive bacteria generally use oligopeptide-type molecules as signaling factors [111]. When the autoinducers reach a critical concentration, they can initiate the expression of relevant genes and regulate the biological behavior of bacteria, such as toxin production and biofilm formation [110]. Therefore, the inhibition of intercellular communication by disrupting QS is a promising method for eradicating biofilm. Both Ag and Au have been reported to be employed to treat biofilm infections as anti-QS agents, which have attracted great attention as well among metal nanomaterials [112,113]. Ghaffarlou et al. developed Ag NPs-decorated Pullulan (Ag NPs@Pull) and Au NPs-decorated Pullulan (Au NPs@Pull) to combat the bacterial resistance (Figure 11a). Ag NPs@Pull and Au NPs@Pull significantly inhibited bacterial signal molecules, according to the results [114]. Unfortunately, Ag NPs are widely known to be extremely hazardous to mammalian cells. Thus, developing an approach that modulates the interaction of QS signals with nanomaterials for reducing the dose of nanomaterials is necessary. Prateeksha et al. established chrysophanol-functionalized biocompatible Ag NPs (CP-Ag NPs) to increases the interaction of chrysophanol and bacterial QS signaling. Changes in the expression of virulence gene demonstrated that CP-AgNPs inhibit QS signaling (Figure 11b). Additionally, CP-Ag NPs were able to disrupt QS signaling to reduce bacterial adhesion and biofilm formation compared to citrate-capped Ag NPs (Cc-AgNPs) [115].

### 4.3. Bacteria

Disruption of the breaking of EPS and QS contribute to biofilm removal. Notably, planktonic bacteria is a prerequisite for biofilm formation and killing the bacteria within biofilm is essential for efficient biofilm removal [118]. Otherwise, the surviving bacteria will form a new biofilm by choosing a suitable substrate to colonize again, causing a more serious infection [119]. Metal-based nanozymes catalyze the conversion of H_2_O_2_ to the more toxic ROS through typical peroxidase-like reaction, which have attracted tremendous attention in antibiofilm therapy [120,121,122]. For example, Kumar et al. eradicated biofilm using mixed FeCo-oxide-based surface-textured nanostructures with magnetocatalytic activity [116]. The findings demonstrated that the biofilm and embedded bacteria were damaged by the defensive ROS (Figure 11c). In addition, the functionalized nanoenzyme can also selectively identify and kill harmful flora without affecting normal flora. Liu et al. reported the application of ferumoxytol iron oxide nanoparticles (FerIONP) for the removal of pathogenic dental biofilms [117]. By catalytically activating H_2_O_2_ to promote in situ free radical generation, FerIONP exhibited remarkable antibacterial effects. At the same time, FerIONP showed great pathogen selectivity by interacting with pathogen-specific dextran-binding proteins, which had no effect on commensal *streptococci* (Figure 11d,e). The activation of the immune system prevents the recurrence of biofilm. Wang et al. reported nitrosothiol-coated CoFe_2_O_4_@MnFe_2_O_4_ nanoparticles as highly efficient antibiofilm platforms [118]. These systems not only removed the presence of biofilm by generating heat and releasing nitric oxide, but also triggered macrophage-related immunity to prevent the recurrence of biofilm infections (Figure 11f).

## 5. Conclusions and Future Perspectives

The latest applications and mechanisms of metal and metal oxide nanomaterials for antibacterial and antibiofilm purposes are reviewed in this paper. In terms of antibacterial application, metal and metal oxides nanomaterials can induce chemical and physical damage to eliminate bacteria, including disrupting the cell membrane potential and generating ROS. Protein is the bearer of life activities. Metal and metal oxides nanomaterials can prevent protein synthesis by disrupting ribosomes or cause the inactivation of important enzymes of bacteria by inducing hyperthermia to disrupt the structure of protein. In addition, copper-based nanomaterials can disrupt the activity of GR to cause the dysregulation of the bacterial antioxidant homeostatic system, ultimately inducing apoptosis of bacterial cells. Nucleic acid is one of the most basic substances of life. Metal and metal oxide nanomaterials can regulate the expression of relative genes to reduce the virulence of bacteria and interfere with bacterial metabolism such as TCA, ultimately leading to bacterial death. For biofilms, metal and metal oxide nanomaterials can disrupt EPS by mechanical force and generating ROS, and they can also assist in disrupting biofilms by suppressing QS. In addition, they can produce ROS to remove bacteria within biofilm or activate the body’s immune system to prevent the recurrence of biofilm.

Compared to conventional antibiotics, metal and metal oxide nanomaterials present unique advantages in the anti-infective field, such as being highly effective and avoiding the development of drug-resistant bacteria. However, there are still many issues to be considered when metal and metal oxide nanomaterials are used in the clinic. The following outlines several important issues that should be considered for future applications from our perspective.

Toxicity issues. The toxicity of metal and metal oxide nanomaterials should not be disregarded, despite the fact that they offer outstanding anti-infection properties, such as Ag nanomaterials. Hydrogels are crosslinked polymers that have outstanding biocompatibility and can hold a lot of water in solution without dissolving. It can be modified to provide a slow and controlled release of the load, such as temperature-sensitive hydrogels. Therefore, the encapsulation of metal and metal oxide nanomaterials in hydrogels to control their release behavior will avoid the toxic side effects of sudden release.Metabolic issues. A further issue is the metabolism of metal ions produced in vivo by the breakdown of metal nanomaterials. Fe, Cu, Zn, and Mo are essential trace elements for the human body. They show excellent antibacterial properties when formed as nanomaterials. For example, iron-based nanomaterials and copper-based nanomaterials present enzyme-like activities that can convert H_2_O_2_ into more toxic hydroxyl radicals for removing bacteria from the wound area. Thus, these nanomaterials should be given priority in the selection of anti-infective materials.Stability issues. Metal and metal oxide nanomaterials are usually unstable and prone to aggregation, especially in small sizes. However, it has been demonstrated that small-sized nanomaterials have a higher likelihood of passing through the membranes of bacterial cells and biofilms. As a result, the aggregation of metal and metal oxide nanomaterials will reduce their efficacy and even increase the toxic side effects. The nanocomposite created by combining metal and metal oxide nanostructures with other nanomaterials such as graphene would be a good choice. This approach not only ensures the dispersion of metal and metal oxide nanomaterials, but also enhances the anti-infective efficacy. In addition, performing ligand modification or wrapping organic layers such as PLGA is also a proper solution.Efficacy issues. The efficacy of metal and metal oxide nanomaterials that work with hyperthermia or ROS is limited by the distance of action. Fortunately, the distance can be reduced by the surface modification of metal and metal oxide material, such as ligand modification and increased surface roughness. This improvement will maximize the anti-infective properties of the nanomaterials while simultaneously maintaining biosafety.Mechanistic explanation. The currently available anti-infective mechanisms lack integrity, which are mostly at the cellular and protein levels. There is an urgent need to construct a complete and systematic anti-infective mechanism of metal and metal oxide nanomaterials based on the gene level, protein level, and cellular level. For example, researchers can study the expression of related genes and proteins from the signaling pathways related to the energy metabolism in the bacterium and explore the effect of materials on the energy production of the bacterium. In addition, researchers can study the redox homeostasis system of the bacterium and elaborate the mechanism of disruption of the antioxidant system of the bacterium by the material to lay the foundation for the design of the material.

In conclusion, the design of a metal-based nanomaterial with efficient anti-infective properties and good biosafety is necessary for the clinical translation of an anti-infective strategy. We believe that this progress report will provide new insights into the mechanisms of action of metal and metal oxide nanomaterials in anti-infective applications and offer new ideas for the development of nanomaterials suitable for clinical applications.

## Figures and Tables

**Figure 1 ijms-23-11348-f001:**
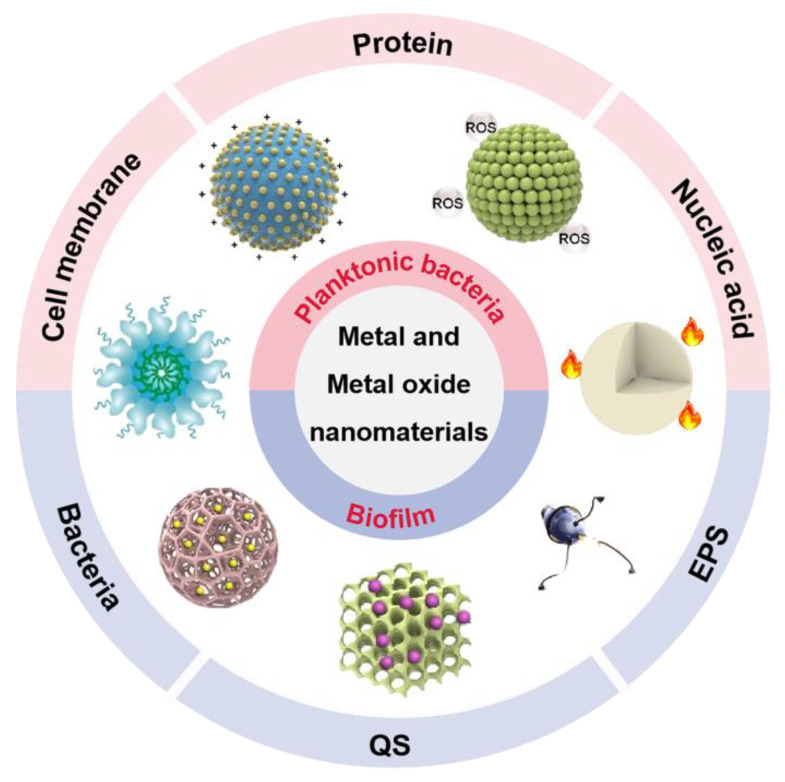
The antibacterial and antibiofilm application of metal and metal oxide nanomaterials. EPS: extracellular polymeric substance; QS: quorum sensing; ROS: reactive oxygen species.

**Figure 2 ijms-23-11348-f002:**
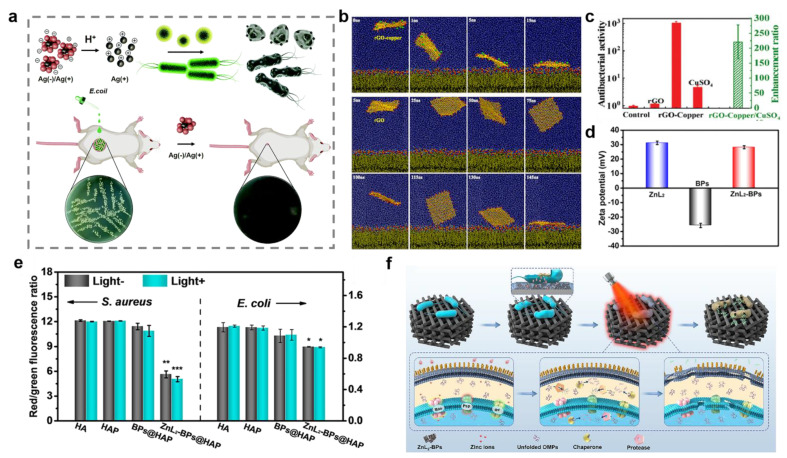
Disruption of bacterial cell membranes through electrostatic interactions. (**a**) Schematic illustration for bacterial killing by acid-responsive Ag (−)/Ag (+) clusters. (**b**) Trajectory of binding between cell membrane and reduced graphene oxide (rGO)-copper composite and rGO. (**c**) Antibacterial activity (**left** axis) of rGO-copper composite and enhancement ratio (**right** axis) compared to copper ions. (**d**) Zeta potentials of zinc sulfonate ligand (ZnL_2_), black phosphorus nanosheets (BPs), and ZnL_2_-BPs. (**e**) Bacterial membrane potentials. * denotes *p* < 0.05, ** denotes *p* < 0.01, and *** denotes *p* < 0.001 compared with the hydroxylapatite (HA) group without near infrared (NIR)-irradiation. (**f**) The antibacterial mechanism of ZnL_2_-BPs@HAP. (**d**–**f**) Reprinted with permission from Ref [71]. Copyright 2021 American Chemical Society.

**Figure 3 ijms-23-11348-f003:**
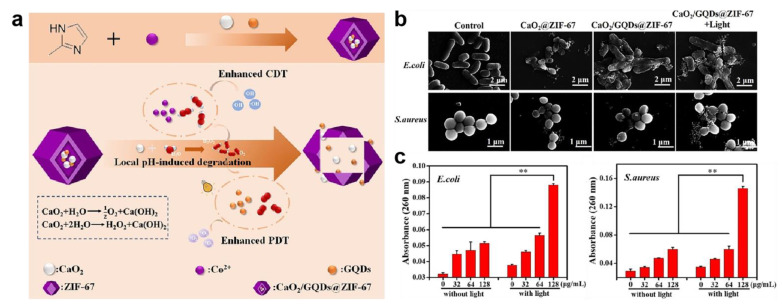
Cell membrane damage caused by zeolitic imidazolate framework-67 nanosystem (CaO_2_/GQDs@ZIF-67). (**a**) Schematic illustration for bacterial killing by CaO_2_/GQDs@ZIF-67. (**b**) Scanning electron microscope image of *E. coli* and *S. aureus*. (**c**) Nucleic acid leaking detection of *E. coli* and *S. aureus* after CaO_2_/GQDs@ZIF-67 treatment. ** denotes *p* < 0.01. Reprinted with permission from Ref [73]. Copyright 2021 American Chemical Society.

**Figure 5 ijms-23-11348-f005:**
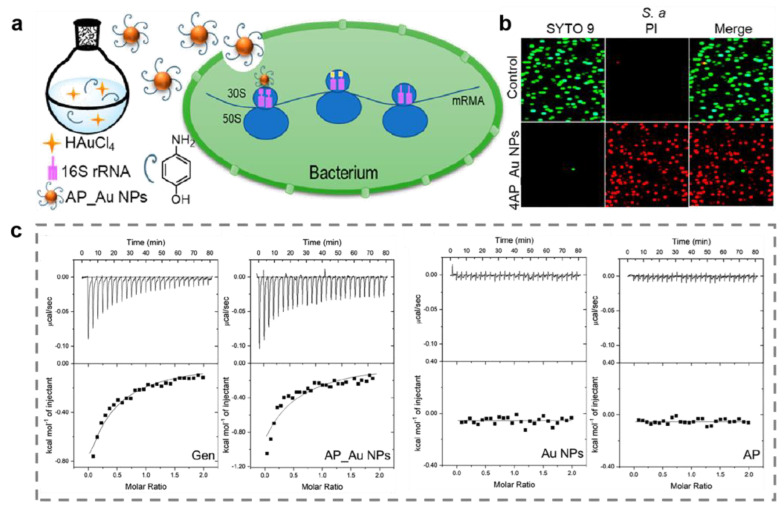
Protein destruction by aminophenol (AP)-modified gold nanoparticles (AP_Au NPs). (**a**) The antibacterial mechanism of AP_Au NPs. (**b**) Evaluation of live-dead staining of bacteria. (**c**) Isothermal titration calorimetry analysis of the interface between different agents and bacterial 16S rRNA. Reprinted with permission from Ref. [81]. Copyright 2022 American Chemical Society.

**Figure 6 ijms-23-11348-f006:**
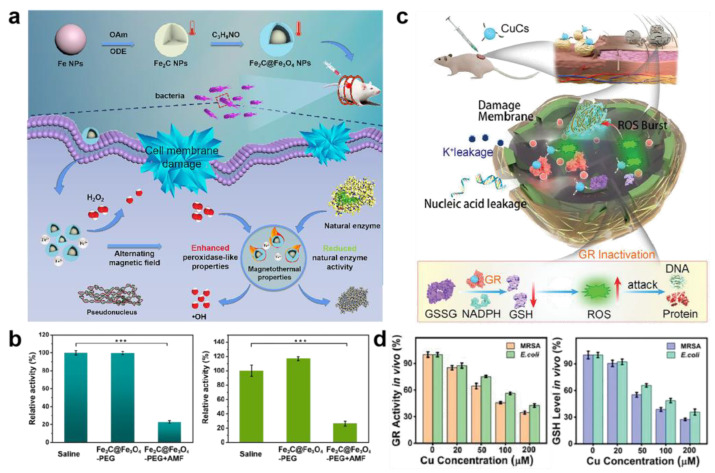
Protein destruction by metal oxide nanomaterials. (**a**) Schematic illustration of preparation process and antibacterial mechanism of Fe_2_C@Fe_3_O_4_-PEG thermogenic nanozyme. (**b**) The activity of glutathione reductase (GR, **left**) and adenosine triphosphatases (**right**) after different treatments. *** indicate obvious differences at *p* < 0.001. (**c**) Scheme of copper clusters (CuCs) healing local *MRSA* skin wound infection. (**d**) Enzymatic activity of GR and reduced glutathione levels in *MRSA* and *E. coli* after treatment with CuCs. GSSG, glutathione disulfide; GSH, reduced glutathione; ROS, reactive oxygen species; NADPH, nicotinamide adenine dinucleotide phosphate.

**Figure 7 ijms-23-11348-f007:**
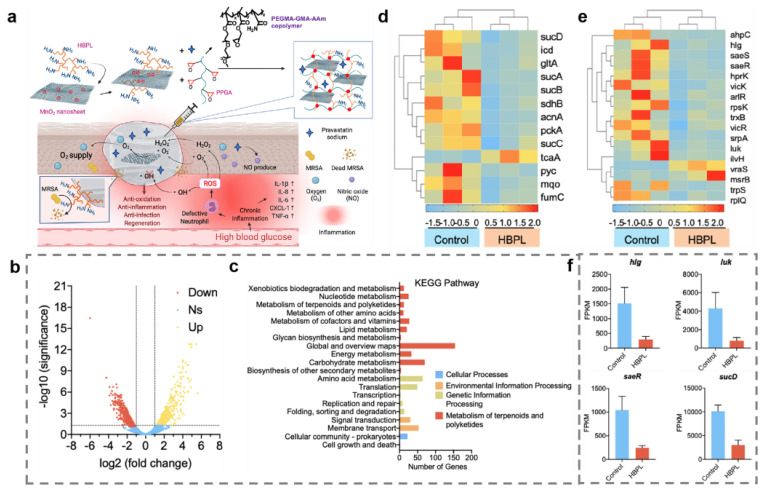
Disruption of bacterial nucleic acids by hyperbranched poly-L-lysine (HBPL)-crosslinked hydrogel containing MnO_2_. (**a**) Antibacterial and wound healing promoting mechanism of HBPL-crosslinked hydrogel containing MnO_2_. (**b**) Volcano plot analyses of total Differentially expressed gene. (**c**) KEGG analyses of DEGs. Heatmap of genes associated with tricarboxylic acid cycle (TCA) (**d**) and virulence (**e**). (**f**) Fragments per kilobase million (FPKM) of *hlg*, *luk*, *saeR*, and *sucD* of *MRSA*.

**Figure 8 ijms-23-11348-f008:**
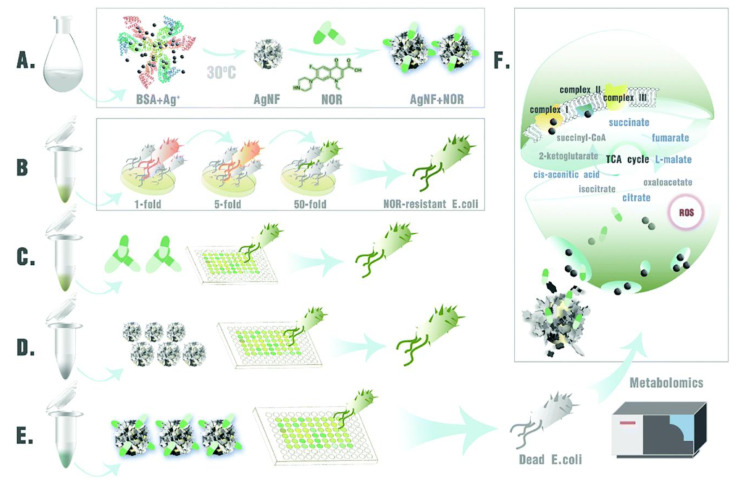
Schematic illustration of the antibacterial mechanism of antibiotic-loaded silver nano flowers (Ag NFs). (**A**) Synthesis of Ag NFs. (**B**) Evolutionary process of NOR-resistant strains. (**C**) Norfloxacin (NOR) alone does not eradicate NOR-resistant *E. coli*. (**D**) Ag NFs alone cannot eradicate NOR-resistant *E. coli*. (**E**) Combination of Ag NFs and NOR eradicates NOR-resistant *E. coli*. (**F**) Metabolomic studies revealed that the combination of Ag NFs and antibiotics combined improved the antimicrobial effect through the release of silver ions and stimulation of the TCA cycle (five upregulated metabolites in the TCA cycle are marked in blue).

**Figure 9 ijms-23-11348-f009:**
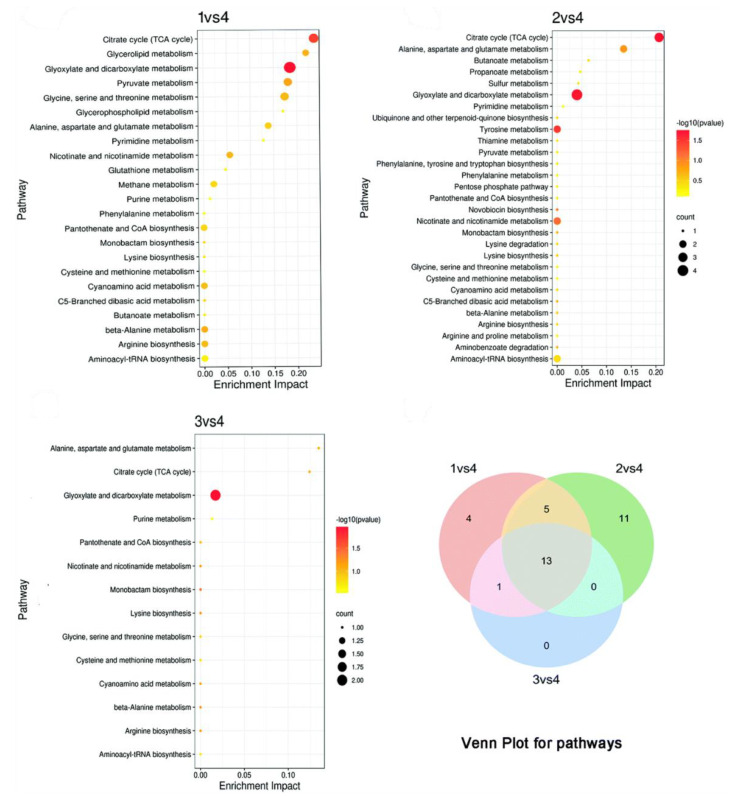
The pathway enrichment and Venn plot of pathways of three group pairs (1: NOR + Ag NFs, 2: Ag NFs, 3: NOR; 4: Control).

**Figure 10 ijms-23-11348-f010:**
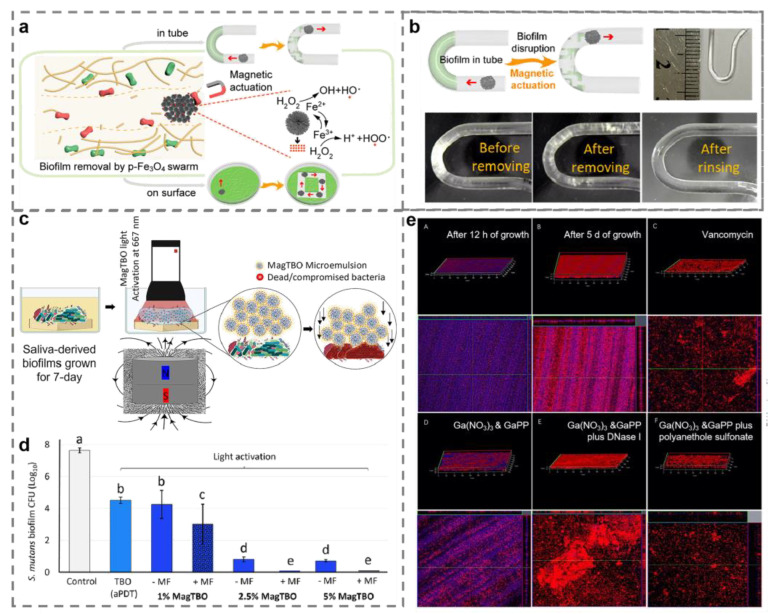
Metal and metal oxide nanomaterials remove biofilm by disrupting extracellular polymeric substance. (**a**) Schematic illustration of biofilm disruption by nanorobots. (**b**) The antibiofilm effect of nanorobots in the U-shaped tube. (**c**) Schematic illustration of biofilm disruption by photosensitizer nanoplatform (MagTBO). (**d**) The antibiofilm effect of MagTBO and toluidine-blue ortho (TBO) alone. Values indicated by different letters are statistically different from each other (*p* < 0.05). (**e**) Environmental DNA (eDNA) alterations in biofilms with different treatment (red represents live bacteria and blue represents eDNA). (**a**,**b**) Reprinted with permission from Ref. [105]. Copyright 2021 American Chemical Society. (**c**,**d**) Reprinted with permission from Ref. [106]. Copyright 2021 American Chemical Society. (**e**) Reprinted with permission from Ref. [109]. Copyright 2021 American Chemical Society.

**Figure 11 ijms-23-11348-f011:**
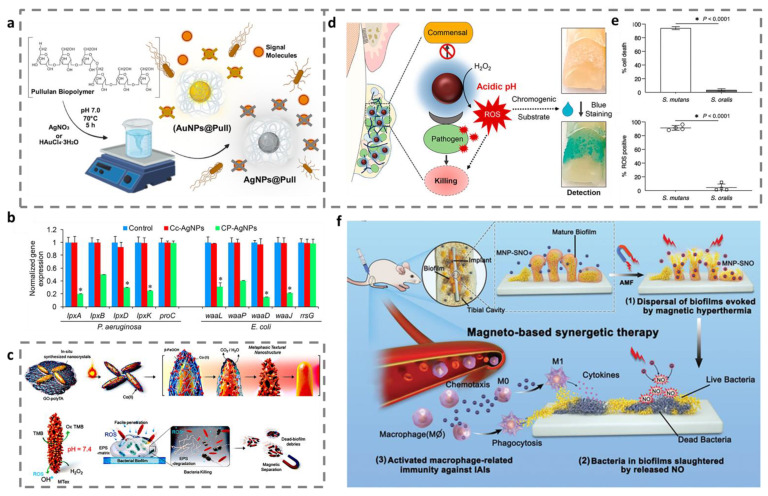
Metal and metal oxide nanomaterials remove biofilm by blocking quorum sensing and killing bacteria. (**a**) The preparation scheme of Ag NPs@Pull and Au NPs@Pull and their quorum sensing inhibition. (**b**) The expression of virulence factor-related genes. * *p* < 0.001 versus control. (**c**) Synthetic scheme and bactericidal effect of mixed FeCo-oxide-based surface-textured nanostructures. (**d**) Schematic diagram of the selective catalytic-therapeutic–diagnostic application of FerIONP. (**e**) Selective bactericidal effect of FerIONP and detection of intracellular reactive oxygen species production. (**f**) Schematic illustration of the antibiofilm mechanism of nitrosothiol-coated CoFe_2_O_4_@MnFe_2_O_4_. (**a**) Reprinted with permission from Ref. [114]. Copyright 2022 American Chemical Society. (**b**) Reprinted with permission from Ref. [115]. Copyright 2021 American Chemical Society. (**c**) Reprinted with permission from Ref. [116]. Copyright 2021 American Chemical Society. (**d**,**e**) Reprinted with permission from Ref. [117]. Copyright 2021 American Chemical Society. (**f**) Reprinted with permission from Ref [118]. Copyright 2021 American Chemical Society.

**Table 1 ijms-23-11348-t001:** The antibacterial mechanism and synthetic method of metal nanomaterials.

Nanomaterials	Synthetic Method	Mode of Action	Bacterial Species	Ref.
Ag NPs	Chemical reduction	Ag^+^	*E. coli* and *S. aureus*	[33]
Nano-Ag	Photoreduction	Ag^+^ and ROS	*E. coli*	[34]
Ag NPs	Chemical reduction	ROS	Carbapenem-resistant *K. pneumoniae*	[35]
Ag NPs	Hydrothermal reaction	Ag^+^	*E. coli*, *S. aureus*, and *C. albicans*	[36]
Au NPs	Seed-mediated growth	ROS	*B. subtilis* and *E. coli*	[37]
Au NPs	Chemical reduction	No information	*E. coli*	[28]
Au NCs	Chemical reduction	ROS	*E. coli*, *S. aureus*, MDR *E. coli*, and MDR *S. aureus*	[29]
Au NRs	Chemical reduction	ROS	*E. coli* and *S. aureus*	[38]
Cu NPs	Solution casting method	Free radicals	*E. coli* and *S. aureus*	[39]
Cu NPs	Atmosphere arc discharge method	ROS	*S. sanguinis*, *P. gingivalis*, and *S. mutans*	[40]
Cu NPs	Biosynthetic method	ROS	*E. coli* and *S. aureus*	[41]

NPs, nanoparticles; NCs, nanoclusters ROS, reactive oxygen species.

**Table 2 ijms-23-11348-t002:** The antibacterial mechanism and synthetic method of metal oxide nanomaterials.

Nanomaterials	Synthetic Method	Mode of Action	Bacterial Species	Ref.
ZnO NPs	Atmosphere arc discharge method	ROS	*S. sanguinis*, *P. gingivalis*, and *S. mutans*	[40]
ZnO NPs	Biosynthetic method	ROS and Zn^+^	*E. coli* and *S. aureus* *P. aeruginosa* and *C. albicans*	[47]
Flower-shaped ZnO	Wet chemical method	ROS	*E. coli*	[48]
Nano-TiO_2_	No information	ROS	*E. coli* and *A. hydrophila*	[49]
TiO_2_ nanowire	Hydrothermal reaction	ROS	Gram-positive bacteria and Gram-negative bacteria	[50]
TiO_2_ NPs	Purchased from Nanostructured and Amorphous Materials Inc. and MK Impex Corp., Division MK Nano	ROS	*E. coli*	[51]

## Data Availability

Not applicable.

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
