# Peer review of "Metal and Metal Oxide Nanomaterials for Fighting Planktonic Bacteria and Biofilms: A Review Emphasizing on Mechanistic Aspects"

_ijms, 2022, doi:10.3390/ijms231911348_

Round 1

Reviewer 1 Report

In this review the authors explain the state-of-the-art applications of metal and metal oxide nanomaterials in bacterial infective disease.

The topic  is interesting . However, some issues should be addressed.

1)    In the introduction, the authors should clearly highlight the novelty of their work in comparison to previous works.

2)      The authors could divide the figures into multiple sub-figures to help readers understand.

Reviewer 2 Report

The review entitled ‘Metal and Metal Oxide Nanomaterials for Fighting Planktonic Bacteria and Biofilms: A Review Emphasizing on Mechanistic Aspects by sun et al., describes how nanomaterial can acts on bacteria. The review is well written and will be useful for the researches. However, before accepting the manuscript I have few suggestions to the authors.

1.       Authors should add a separate section for the different kinds of metal nanoparticles  and their mode of actions on different species of the bacteria for example lot of work has been done from past decade about the silver nanoparticles, Zinc, Gold, nickel etc . Authors should add separate sections for metallic nanoparticles.

2.       Authors should also add different methods used to synthesize metal based nanoparticles.

The manuscript should be accepted after minor revision.

Author Response

Point 1: Authors should add a separate section for the different kinds of metal nanoparticles and their mode of actions on different species of the bacteria for example lot of work has been done from past decade about the silver nanoparticles, Zinc, Gold, nickel etc. Authors should add separate sections for metallic nanoparticles.

Response 1: Many thanks for your helpful comment. According to your advice, we summarize some common metal nanomaterials and metal oxide nanomaterials that have been used for anti-infective purposes in recent years. The relevant paragraph has been replaced as:

Metal and metal oxide nanomaterials refer to metals and alloys that form nanograins with small size effect, quantum effect, surface effect and interface effect. They have unique physical and chemical properties compared to traditional metal and metal oxide materials, which have been widely investigated in recent years in different fields. Notably, they have been widely used in the anti-infective field due to their excellent anti-infective properties.

2.1. Metal Nanomaterials

Metal nanomaterials have been widely used in the field of anti-infection due to their excellent anti-infective efficacy. Among them, Ag, Au and Cu are the most common anti-infective nanomaterial (Table 1). For thousands of years, the antibacterial properties of Ag have been discovered and used in everyday life, such as the use of silverware. The antibacterial properties of Ag nanomaterials mainly derive from Ag+ and depend largely on their size and shape. For example, Yang et al. used the displacement reaction between Zn and Ag+ to introduce Ag nanomaterials into the metal-organic framework. The antibacterial mechanism showed that massive release of Ag+ destroyed the bacterial contents and enhanced the effectiveness of the nanocomposite against S. aureus and E. coli[25]. Different from Ag nanomaterials, the antibacterial properties of Au nanomaterials are largely affected by their morphology and the antibacterial mechanisms are diverse. For example, Au NPs lack antibacterial activity on their own, but they show favorable antibacterial activity by surface modification[26,27]. Based on this, Li et al. used 4,6-diamino-2-pyrimidinethiol modified Au NPs for the treatment of infections caused by E. coli[28]. In contrast, Au nanoclusters (NCs) with same surface ligand exhibit broad-spectrum antibacterial property by inducing over-accumulation of ROS, not only against Gram-negative and Gram-positive bacteria, but also against their multidrug-resistant bacteria[29]. As another common anti-infective nanomaterials, Cu nanomaterials perform antibacterial properties mainly through generating ROS. On the one hand, Cu destroy the bacterial antioxidant system by causing the inactivation of GR, resulting in a surge of ROS[30]. On the other hand, cuprous ions originate from copper-based nanomaterials generated ROS due to Fenton-like activity. For example, Lin et al. constructed copper ion-loaded melanin and copper ion-loaded polydopamine to treat infective diseases induced by S. aureus and E. coli with the help of copper ion release and copper ion-induced ROS production[31]. Besides, Pd and Pt nanomaterials can also act as nanozymes to produce ROS, resulting in broad-spectrum antibacterial activity against both Gram-negative and Gram-positive bacteria[32].

Table R1 The antibacterial mechanism and synthetic method of metal nanomaterials.

Nanomaterials

Synthetic method

Mode of action

Bacterial species

Refs

Ag NPs

Chemical reduction

Ag+

E. coli and S. aureus.

[33]

Nano-Ag

Photo-reduction

Ag+ and ROS

E. coli.

[34]

Ag NPs

Chemical reduction

ROS

Carbapenem-resistant K. pneumoniae.

[35]

Ag NPs

Hydrothermal reaction

Ag+

E. coli, S. aureus and C. albicans.

[36]

Au NPs

Seed-mediated growth

ROS

B. subtilis and

E. coli.

[37]

Au NPs

Chemical reduction

No information

E. coli.

[28]

Au NCs

Chemical reduction

ROS

E. coli, S. aureus, MDR E. coli and MDR S. aureus.

[29]

Au NRs

Chemical reduction

ROS

E. coli and S. aureus.

[38]

Cu NPs

Solution casting method

Free radicals

E. coli and S. aureus.

[39]

Cu NPs

Atmosphere arc discharge method

ROS

S. sanguinis, P. gingivalis and S. mutans

[40]

Cu NPs

Biosynthetic method

ROS

E. coli and S. aureus.

[41]

2.2. Metal Oxide Nanomaterials

Compared to metal nanomaterials, metal oxide nanomaterials have attracted the highest interest in the anti-infective community due to the better biological properties, such as TiO2 and ZnO (Table R2)[42]. As semiconductor nanomaterials, TiO2 and ZnO can generate highly toxic ROS by photocatalytic property, which are viewed as the promising tool for anti-infection therapy. However, their photocatalytic-based antibacterial properties were limited to the narrow response range of visible light and easy recombination properties of photo-induced electron-hole pairs[43]. In order to improve their photocatalytic performance, combining with a semiconductor featuring narrow band gap have been reported. For example, Khan et al. reduced band gap using Ag2S and decreased the rate of recombination for photoinduced charge carriers using graphene oxide (GO)[44]. The obtained Ag2S-ZnO/GO nanocomposite showed outstanding photocatalytic property and remarkable antibacterial activity than pure ZnO nanomaterials. Besides, ZnO nanomaterials can also exert antibacterial properties through contact adsorption mechanism and metal ion dissolution mechanism. They all act by zinc ions derived from the degraded of ZnO nanomaterials in acidic environment[45]. Zinc ions cause membrane potential disruption by adhering to the cell membrane. Besides, they also act on the thiol group of the bacterial respiratory enzymes to increase the production of ROS, ultimately leading to bacterial death[46].

Table R2 The antibacterial mechanism and synthetic method of metal oxide nanomaterials.

Nanomaterials

Synthetic method

Mode of action

Bacterial species

Refs

ZnO NPs

Atmosphere arc discharge method

ROS

S. sanguinis,

P. gingivalis and

S. mutans

[40]

ZnO NPs

Biosynthetic method

ROS and Zn+

E. coli, S. aureus,

P. aeruginosa and

C. albicans.

[47]

Flower-shaped ZnO

Wet chemical method

ROS

E. coli

[48]

Nano-TiO2

No information

ROS

E. coli and

A. hydrophila

[49]

TiO2 nanowire

Hydrothermal reaction

ROS

Gram-positive bacteria and Gram-negative bacteria

[50]

TiO2 NPs

Purches from Nanostructured &

Amorphous Materials Inc. and MK

Impex Corp., Division MK Nano

ROS

E. coli

[51]

Point 2: Authors should also add different methods used to synthesize metal based nanoparticles.

Response 2: Many thanks for your suggestion. According to your advice, we have summarized several common methods used to synthesize metal-based nanomaterials, which are described in Page 5 and Page 6 of revised manuscript as:

2.3 Synthetic method

Different nanomaterials exhibit different degrees of anti-infective effect, which directly depend on their composition, morphology and size. These characteristics are closely related to the synthetic method. There are various methods for the synthesis of metal and metal oxide nanomaterials, including chemical reduction, chemical precipitation, Brust-Schiffrin method, seed-mediated growth, hydrothermal reaction and biosynthetic methods[52,53].

Chemical and biosynthetic methods have been reported for the preparation of metal-based nanomaterials. Among them, chemical reduction is the most common chemical method. However, the use of reducing agents raises the toxicity and cost of the method, as well as introduces impurities. As a simple and fast method, chemical precipitation method has attracted a lot of attention, which allow controlling the size and shape of nanomaterials. The reaction rate and nucleation process are largely affected by reaction parameters such as pH, temperature and reactant concentration. In addition, the size, shape and properties of nanomaterials will depend on the crystallization process. Based on this, Sondi et al. prepared well-dispersed Ag NPs with the size of 12.3 nm using the reduction of AgNO3 by ascorbic acid[54]. Wang et al. synthesized morphologically controllable ZnO NPs nanoparticles with a particle size of about 20 nm using ZnCl2 as precursor and ammonium carbamate as precipitating agent[55]. As a non-toxic and environmentally friendly method, biosynthetic method can be used to synthesize various nanomaterials with a wide range of size, physicochemical properties, shapes, and compositions, which has been widely reported. The sources of nanomaterials include plants, bacteria and algae. For example, Mori et al. reported a simple and environmentally friendly route for the biosynthesis of Ag NPs[56]. The size of Ag NPs was controlled by varying the glucose concentration. The final Ag NPs were produced with a controlled particle size range of 3.48±1.83 ~ 20.0±2.76 nm. Nasrollahzadeh et al. synthesized Cu NPs using Euphorbia grandis leaf extract as a reducing and stabilizing agent without surfactants[52].

Brust-Schiffrin method and seed-mediated growth synthesis method can be employed to produce Au nanomaterials[57]. Brust-Schiffrin is the first special method that can generate thiolate-stabilized Au NPs. The Au NPs synthesized by this method have the following advantages: 1) high thermal and air stability, 2) No aggregation or decomposition occurs during repeated separation and redecomposition, 3) easy adjustment of the particle size of the synthesized gold nanoparticles with narrow dispersion range, 4) relatively easy functionalization and ligand substitution modifications. For example, Selina Beatrice et al. prepared molecular tweezers functionalized ultrafine Au NPs that selectively adsorb to lysine and arginine residues on protein surfaces using the Brust-Schiffrin method[58]. The process of seed-mediated growth synthesis method of Au nanomaterials can be divided into two stages. First, small-sized Au NPs is synthesized as seeds. Second, the seeds are added to a "growth" solution consisting of HAuCl4, stabilizer and reducing agent. Au0 produced by the reduction usually appears on the seed surface and eventually a large amount of Au NPs is formed. Based on this method, MSc et al. obtained gold nanorods, gold nanostars and gold nanospheres with small size and good dispersion[59].

Hydrothermal reaction is a method of synthesis using chemical reactions of substances in aqueous solutions at temperatures of 100 to 1000 °C and pressures of 1 MPa to 1 GPa. It can create new nanocompounds and nanomaterials that cannot be prepared by other methods because the homogeneous nucleation and non-homogeneous nucleation mechanisms of hydrothermal reaction are different from the diffusion mechanisms of solid-phase reactions. Importantly, the products obtained by hydrothermal reaction have high purity, good dispersion and easy particle size control. For example, Wang et al. obtained well-dispersed copper nanowires using CuCl2·2H2O as the copper source and PVP as the dispersant. Ozga et al. used a modified hydrothermal reaction to obtain copper oxide films at less than 101 °C. Li et al. made ZnO nanoflowers by adjusting the ratio of zinc nitrate hexahydrate and cyclic hexamethylenetetramine[60]. Huang et al. prepared TiO2 nanotubes with high catalytic efficiency by hydrothermal reaction.

To sum up, nanomaterials of the same component prepared by different methods may have different properties. It is important to choose an appropriate method by combining various factors.